

# Aerosol effects on day-ahead solar radiation forecasting

Xinyuan Hou[1], Kyriakoula Papachristopoulou[1], and Stelios Kazadzis[1]

[1]Physikalisch-Meteorologisches Observatorium Davos/World Radiation Center (PMOD/WRC)

**Correspondence:** Xinyuan Hou (xinyuan.hou@pmodwrc.ch)

**Abstract.** We used aerosol data from surface-based AErosol RObotic NETwork (AERONET) and day-ahead aerosol optical depth (AOD) forecasts from the Copernicus Atmosphere Monitoring Service (CAMS) to examine the spatiotemporal variations in AOD at selected sites worldwide. We evaluated three methods for day-ahead AOD forecasting: AERONET 1-day persistence or monthly mean, along with CAMS forecast. High values of daily mean AOD indicates larger day-to-day variability in AOD
and lower predictability. Using the radiative transfer model, we quantify deviations in forecasts of clear-sky direct normal irradiance (DNI) induced by errors in AOD forecasts. The performance of each AOD forecast method in DNI forecast is assessed and compared. Taking into account the characteristic aerosol types at selected locations, we also draw quantitative implications about the reliability and usability of CAMS AOD forecasts for DNI forecasts as alternatives to AOD forecasts based on approaches using ground measurements. For example, CAMS forecasts perform better at more sites than AERONET
persistence, among them many urban-industrial aerosol sites. AERONET persistence forecasts AOD with lower errors at dust aerosol sites. To date, none of the forecast methods for AOD discussed here reliably achieve an accuracy of < 5 % deviation in day-ahead DNI forecasts, but most of the sites can expect better DNI forecasts with a threshold of 20 % DNI deviation.

## 1 Introduction

Besides solar photovoltaic (PV), concentrating solar power (CSP) is another promising solar energy technology growing fast
in recent years (IEA, 2020). CSP only operates in regions with high direct normal irradiance (DNI, > 200 $\mathrm{Wm}^{-2}$) and low cloud cover (Schroedter-Homscheidt et al., 2013). Accurate and reliable forecasts of solar resources are important for both PV systems and CSP plants (Yang et al., 2022), which possess the potential to mitigate energy crisis and climate change at the regional and global scales.

Solar forecasts of global irradiance for PV systems are primarily affected by the uncertainty of clouds. DNI, as part of global
irradiance, is attenuated by aerosols to a larger extent than the diffuse component. Therefore, aerosols play the main role in DNI forecasts for CSP applications in regions with high insolation and low cloudiness such as deserts (Xu et al., 2016), where the soiling of solar collectors due to dust is a concern (Yang et al., 2022). The intensity of aerosols critically affects surface solar radiation (SSR) availability in sunshine-privileged regions, including North Africa (Xiong et al., 2020; Neher et al., 2017), Middle East (AL-Rasheedi et al., 2020; Gueymard and Jimenez, 2018), the Mediterranean (Tuna Tuygun and Elbir, 2024;
Masoom et al., 2023; Fountoulakis et al., 2021), or regions suffered from air pollution such as Northern China (Gao et al.,





2024; Tang et al., 2023) and India (Masoom et al., 2021). Even Central Europe belongs to a much-affected region with higher sensitivity of solar energy production to aerosols (Blaga et al., 2024).

The extinction of solar radiation by atmospheric aerosols is conventionally quantified by aerosol optical depth (AOD). Studies show that the disagreement in irradiances between models and measurements is often linked to models' AOD input
(Yang et al., 2022; Gueymard, 2010). To forecast short-term AOD (e.g., within two days ahead) before assessing its effect on DNI, it is essential to understand its temporal variability in the first place. Sources of AOD data can be ground-based measurements (e.g., the AErosol RObotic NETwork, AERONET, Holben et al. (1998), or the Global Atmosphere Watch Precision Filter Radiometer, GAW-PFR Network, Kazadzis et al. (2018)) or satellite observations. On the one hand, ground-based stations measure aerosols more accurately based on passive remote sensing using radiometers and active remote sensing
using LiDARs. However, compared to measurement stations dedicated to other meteorological parameters such as temperature and precipitation, ground sites measuring AOD remain sparse at the global scale (Sengupta et al., 2021). On the other hand, contemporary satellite observations provide vast spatial coverage and long records with relatively high sampling frequency (Gkikas et al., 2021).

The literature contains several studies that investigated the effect of aerosols on solar radiation forecasts: Gueymard (2012)
introduced the Aerosol Variability Index to describe the temporal variability of AOD from daily to yearly scales and the Aerosol Sensitivity Index to quantify the effects of absolute variations in AOD on relative variations in SSR. Schroedter-Homscheidt et al. (2013) examined the DNI deviation induced by deviations in AOD across the globe using ground-based measurements and atmospheric modeling data. They then discussed the usability of AOD products in solar radiation forecasting, especially DNI under clear-sky conditions. Salamalikis et al. (2021) also evaluated the influence of AOD accuracy on uncertainties in cloud-
free DNI estimates using AOD reanalysis products of global coverage. More recently, Chen et al. (2023) classified four aerosol types based on size distributions and absorptivity using AERONET data and determined the influence of aerosol properties on surface aerosol radiative forcing efficiency. Ansari and Ramachandran (2024) compared the aerosol products from CAMS and MERRA-2 in terms of physical properties and spatiotemporal variability over Asia and discovered a superior performance of CAMS in modeling AOD.

However, it remains unclear to which degree we can reconcile the reliability of ground-based measurements of AOD with the wide coverage of model-based AOD for use cases of DNI forecasts worldwide. The questions include which AOD source can provide day-ahead forecasts of irradiance with what level of accuracy and which forecast method to use at a site with certain aerosol types. This study aims to first examine AOD data sets based on ground-based measurements at selected sites worldwide and quantify the day-to-day AOD variation. We then evaluate three methods to forecast AOD. Next, using the radiative transfer
model (RTM) calculations, deviations in clear-sky DNI caused by differences in AOD forecast are quantified, which are directly linked to the accuracy of DNI forecasts. One objective is to assess the reliability and usability of model-based data for AOD forecasts as alternatives to AOD forecasts based on approaches using ground measurements. Last but not least, we also assess DNI forecasts on locations with different aerosol types and draw implications.



## 2    Data

We used the following aerosol data sets: Level-2 (cloud screened and quality assured) ground-based AOD measurements
from the AERONET Version 3 (Giles et al., 2019) and AOD forecasts from the Copernicus Atmosphere Monitoring Service
(CAMS). AERONET AOD measurement is commonly used as the ground truth for validating and assessing satellite retrievals
or reanalysis-based AOD products (Zhang et al., 2024). CAMS obtains the initial conditions of each forecast by combining
a previous forecast with current satellite observations through data assimilation (Bozzo et al., 2020). The aerosol modeling
scheme includes the following components: dust, organic carbon, black carbon, sulfate AOD and sea salt. CAMS forecasts are
validated against ground-based measurements from AERONET and are available from 2015, providing hourly forecasts up to
five days ahead. Validation of CAMS reanalysis with AERONET data (Inness et al., 2019) suggests a mean bias of -0.003 $\pm$
0.110 in total AOD globally for the period 2003-2016, with positive mean biases over North America and Africa, and largest
standard deviation (0.184) over Southeast Asia.

We selected 21 AERONET sites worldwide for the analysis (Fig. 1). Our first criterion of the AERONET site selection
is the length of the records with consecutive days from 2010 to 2020. At most sites selected, more than 1200 daily values
(average calculated from at least three measurements during the day) for consecutive days from 2010 to 2020 are available.
The second criterion follows the aerosol classification by Hamill et al. (2016), which classifies AERONET sites worldwide
according to five major aerosol types: biomass, dust, maritime, mixed and urban-industrial. A wide geographical distribution
is also considered as the third selection criterion. Therefore, the sites with consecutive days of records < 1000 (Beijing, Capo
Verde and Kuwait) are nevertheless included in our analysis. Selected sites with countries, coordinates, number of consecutive
days with available data and representative aerosol types are listed in Table 1.

## 3    Methodology

Assuming AOD is invariant during the day, we used daily AOD (average calculated from at least three measurements during
the day) at the wavelength of 500 nm (AOD500) from AERONET sites as the reference. Hourly forecasts of AOD at 550 nm
(AOD550) on the following day are extracted from CAMS based on the coordinates of the AERONET sites. The Ångström
exponent (AE) between 440 and 870 nm from AERONET is applied in the Ångström formula to interpolate AOD to the
common wavelength 500 nm with AERONET as expressed in Eq. 1:

$$\mathrm{AOD}_{500} = \frac{\mathrm{AOD}_{550}}{\left(\frac{550}{500}\right)^{-\mathrm{AE}}} \tag{1}$$

Day-to-day AOD variation is quantified for each site. To forecast the day-ahead AOD, we examined three approaches:

1. persistence (assumes AOD remains the same on the next day) using AERONET,

2. monthly mean (2010-2020) AOD from AERONET,

3. CAMS AOD forecast product.





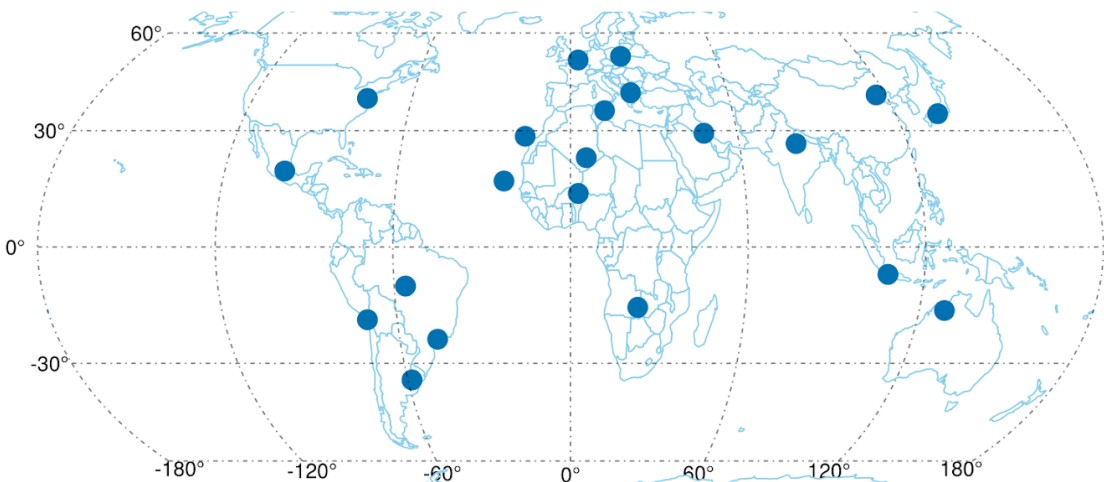

**Figure 1.** Map of 21 selected AERONET sites.

Compared with the day-ahead AOD measurement from AERONET, Pearson correlation coefficients and root-mean-square

error (RMSE), mean absolute error (MAE) and mean bias error (MBE) with standard deviation by each method are computed for each site. Based on these accuracy measures in AOD forecasts compared with AERONET measurements, the optimal forecast method is identified for each site. We also discuss the characteristics of AOD forecasts at different locations with representative aerosol types. The SSR simulation of DNI for cloud-free conditions was performed using the uvspec model from the libRadtran package (Emde et al., 2016; Mayer and Kylling, 2005). Besides AOD, AE and solar zenith angle (SZA),

other input parameters needed are the total column water vapor (TCWV), single scattering albedo (SSA), total ozone column (TOC) and the Earth's albedo. TCWV and SSA are both available from AERONET, where we adopted monthly mean SSA calculated from daily values. We obtained TOC from the Ozone Monitoring Instrument (OMI) TOMS-Like Level-3 product (Bhartia, 2012), which is available daily on a $1° \times 1°$ global grid. Pre-calculated look-up tables (LUT) provide hourly solar irradiance values using combinations of possible parameters: AOD (0:0.05:2, 2.5, 3), AE (0:0.4:2), TCWV (0:1:3, in cm),

SSA (0.6:0.1:1), TOC (200:100:400, in Dobson Unit) and SZA (1:1:89, in °) and the surface albedo was set to 0.2. Relative deviation in DNI caused by deviation in AOD forecast is computed for individual sites. While focusing in more detail on the selected sites with certain aerosol characteristics, we also draw implications at a regional scale. Table 2 provides an overview of the used datasets.

To take into account the diurnal variability of AOD, we compared the effect of using daily or hourly AOD forecasts by

CAMS on simulated DNI for the site Beijing, which has the highest AOD variability among the selected sites. Intra-hour AOD measurements from AERONET are assigned timestamps of the closest hour to match the hourly AOD forecasts from CAMS. Next, we computed daily integrals of DNI estimates based on AOD by three forecast methods and other parameters listed in



**Table 1.** Information on the stations (alphabetically ordered) from the AERONET used in this study. N refers to the number of quality-assured consecutive days at each site.

| Site | Country | lat. [°] | lon. | N | Aerosol type |
|------|---------|----------|------|---|--------------|
| Alta Floresta | Brazil | 9.871 S | 56.104 W | 1718 | B(iomass) |
| Arica | Chile | 18.472 S | 70.313 W | 2191 | U(rban industrial), Mi(xed) |
| Bandung | Indonesia | 6.9 S | 107.6 E | 1522 | Mi |
| Banizoumbou | Niger | 13.547 N | 2.665 E | 2839 | D(ust) |
| Beijing | China | 39.977 N | 116.381 E | 739 | Mi |
| Belsk | Poland | 51.837 N | 20.792 E | 1611 | U |
| Capo Verde | Capo Verde | 16.733 N | 22.935 W | 534 | D |
| CEILAP-BA (Buenos Aires) | Argentina | 34.555 S | 58.506 W | 2277 | B, Mi |
| GSFC (Washington D.C.) | USA | 38.992 N | 76.84 E | 2810 | U |
| Kanpur | India | 26.513 N | 80.232 E | 2647 | Mi, D |
| Kuwait(_Uni) | Kuwait | 29.3 N | 48.0 E | 600 | D |
| Lake_Argyle | Australia | 16.1 S | 128.7 E | 2170 | B |
| Lampedusa | Italy | 35.517 N | 12.632 E | 1569 | Ma(ritime), D |
| Lille | France | 50.612 N | 3.142 E | 1965 | U |
| Mexico City | Mexico | 19.334 N | 99.182 W | 2061 | U, B, Mi |
| Mongu(_Inn) | Zambia | 15.3 S | 23.1 E | 1558 | B |
| Osaka | Japan | 34.651 N | 135.6 E | 2216 | Mi |
| Santa Cruz Tenerife | Spain | 28.473 N | 16.247 W | 2850 | Ma, D |
| Sao Paulo | Brazil | 23.561 S | 46.735 W | 1237 | U, Mi |
| Tamanrasset(_Inn) | Algeria | 22.790 N | 5.53 E | 2728 | D |
| Thessaloniki | Greece | 40.630 N | 22.96 E | 2017 | U |

**Table 2.** Overview of the used datasets.

| Data source | Parameter | Spatial resolution | Temporal coverage | Reference |
|-------------|-----------|--------------------|--------------------|-----------|
| AERONET | AOD, AE, WV, SSA | by site | varies by sites | Giles et al. (2019) |
| CAMS forecast | AOD | $0.4° \times 0.4°$ | hourly since 2015 | Bozzo et al. (2020) |
| OMI TOMS-Like | $O_3$ | $1° \times 1°$ | daily since 2004-10-01 | Bhartia (2012) |





Table 2, before calculating the percentage of days with predefined thresholds of DNI deviation compared with simulated DNI using AOD measurements from AERONET.

## 4    Results and discussion

We first present the results of daily AOD at 500 nm at the 21 selected AERONET sites. Figure 2 shows the distribution of daily AOD for all the sites grouped by regions. In general, European and American sites have the lowest mean AOD, as found in Papachristopoulou et al. (2022). The majority of the sites have its 3rd quartile lower than 0.5. Kanpur, an Indian site characterized by mixed and dust aerosols, has the highest AOD median, partly because South Asia is heavily influenced by the coarse mode dust aerosol from seasonal transport (Ansari and Ramachandran, 2024). Also a mixed aerosol site, Beijing has the largest interquartile range (IQR) in daily AOD.

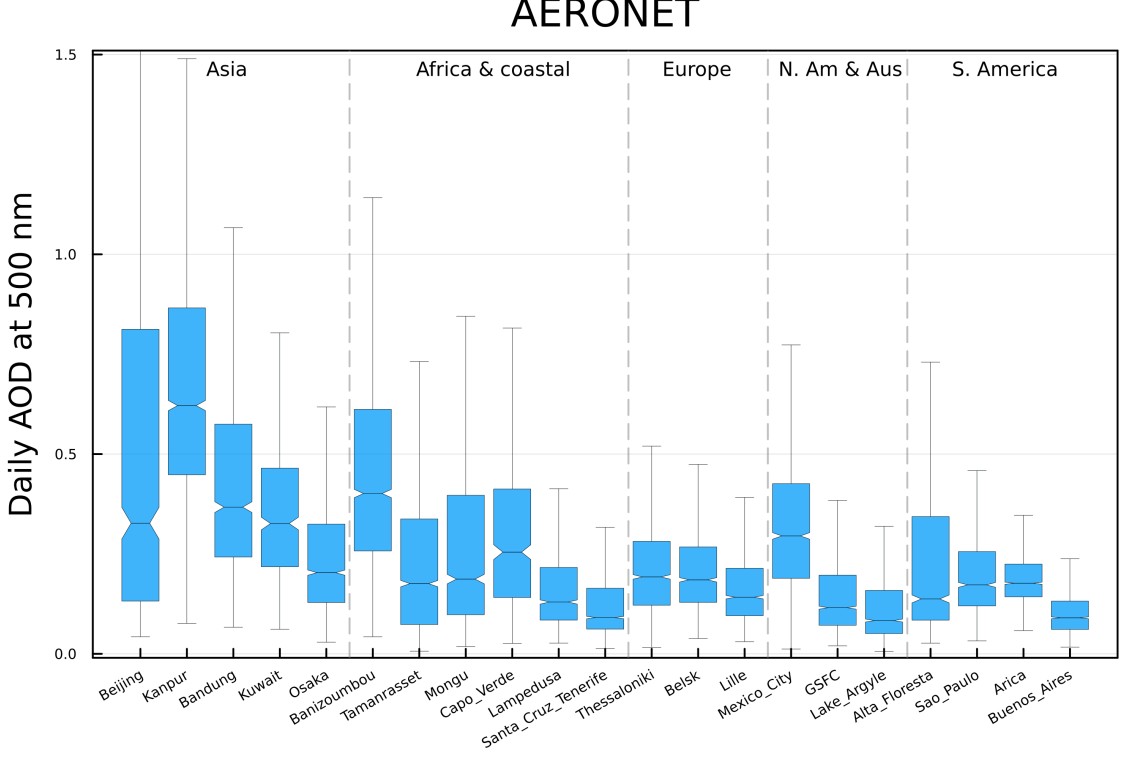

**Figure 2.** Distribution of daily AOD at 500 nm for 21 AERONET sites in this study. Boxes expand the interquartile range (IQR) of the differences. Whiskers correspond to 1.5 times the IQR. Outliers are not plotted. For readability, we set the y-axis limit to be 1.5, which cut the upper whisker of the Beijing box.

Dust aerosol-dominated sites such as Banizoumbou and Tamanrasset in Northern Africa, as well as Kuwait in Middle East generally have over-average high AOD values. The mixed aerosol site Bandung (Indonesia) is also among the sites with the



highest daily AOD. Lampedusa and Santa Cruz de Tenerife, both islands near the African coast, belong to maritime aerosol
sites and have lower daily AOD than dust sites.

At the three selected European sites (Belsk, Lille and Thessaloniki), all of them characterized by urban-industrial aerosols,
the IQR of daily AOD is similar. In Japan, a significant amount of urban-industrial aerosols exists (Hamill et al., 2016), as
the site Osaka exemplifies. Another urban site, Arica (Chile), has the smallest IQR in daily AOD among all selected sites.
Compared to Arica, the site GSFC (Goddard Space Flight Center, situated in suburban Washington, D.C., USA) has a lower
limit on daily AOD. There, local emissions are dominated by automobiles rather than industry (Smirnov et al., 2002).

Sites with biomass aerosols Alta Floresta in the Amazonia, Lake Argyle (Australia) and Mongu in Southern Africa share a
similar pattern, with the range of the 3rd quartile much larger than the 2nd one. The Southern American sites Buenos Aires and
São Paulo both have considerable amount of mixed aerosols, yet Buenos Aires has overall the lowest AOD among the selected
sites.

## 4.1   Day-to-day AOD variability

The distribution of absolute day-to-day differences in AOD (Fig. 3) for the selected sites shares a similar pattern to the distri-
bution of mean daily AOD (Fig. 2). Beijing is the site with the largest day-to-day AOD variability among them, exceeding one
at the upper limit. The day-to-day AOD variability is sufficiently close between Bandung and Kanpur, both among the highest.
Mexico City also has a higher than average day-to-day variation in AOD. For these aforementioned sites, the proportion of
mixed aerosols is considerable. On the other side, Arica, Buenos Aires and Lake Argyle have the smallest day-to-day AOD
variability, with the IQR smaller than other sites. Sites with day-to-day AOD variability on the lower end (the 3rd quartile or
median < 0.1) further include Alta Floresta, Belsk, GSFC, Lampedusa and Santa Cruz de Tenerife. Therefore, sites classified
as predominantly biomass aerosols, maritime aerosols and some urban-industrial aerosol sites have lower day-to-day AOD
variability than sites with other major aerosol types.

The monthly distribution of absolute day-to-day difference in AOD for three selected sites is shown in Fig. 4. Alta Floresta is
characterized by drastically increased aerosol load from September, which could be associated with seasonal biomass burning
in Amazonia (Schumacher and Setzer, 2024). Arica, situated on the northwestern Chilean coast, has a low day-to-day AOD
variation throughout the year (also low seasonal variability) despite its arid desert climate. Beijing, as mentioned earlier, has
relatively high day-to-day AOD variation all year round, although most pronounced during summer. In addition, anthropogenic
emissions in autumn and winter result in frequent severe haze events in Beijing, significantly reducing available SSR there
(Cheng et al., 2022).



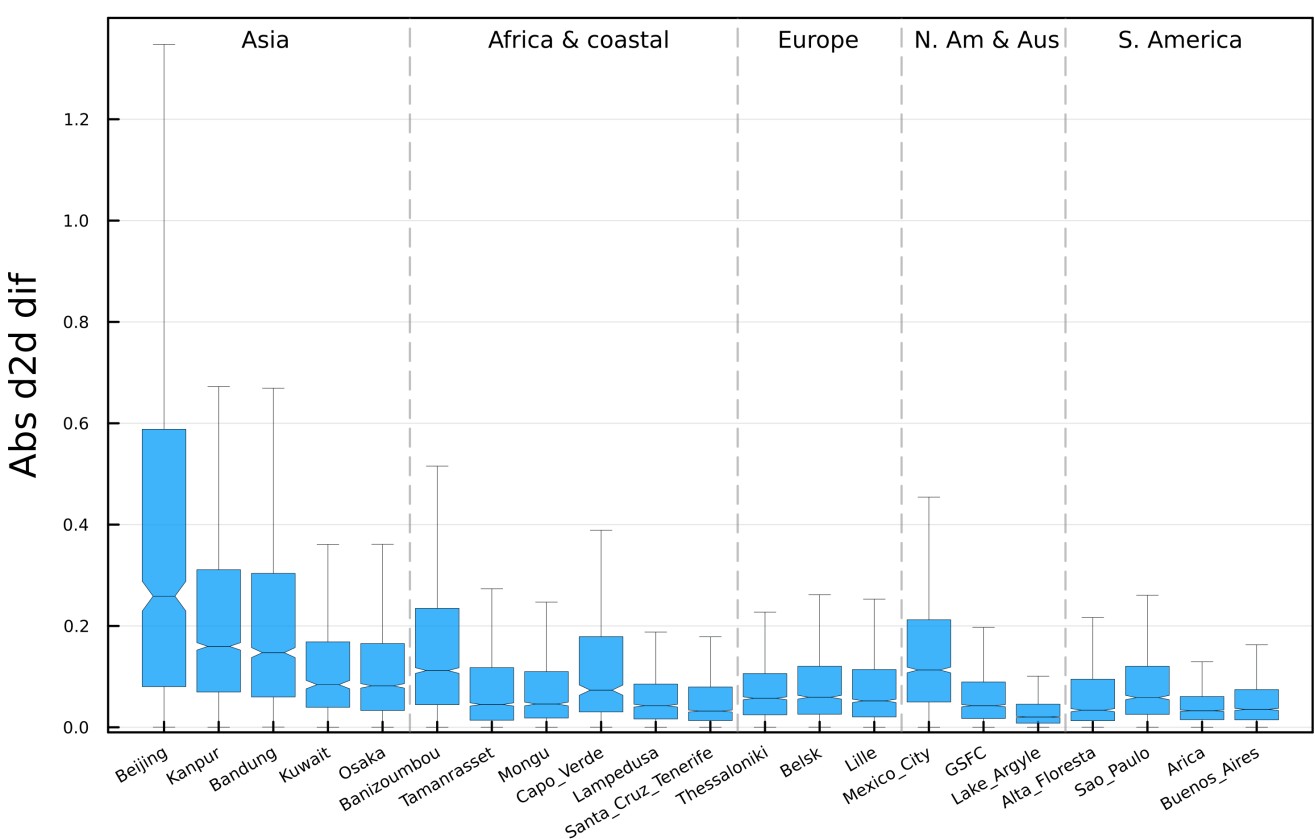

**Figure 3.** Distribution of absolute day-to-day difference in daily mean AOD for 21 AERONET sites in this study. Boxes expand the interquartile range (IQR) of the differences. Whiskers correspond to 1.5 times the IQR. Outliers are not plotted.

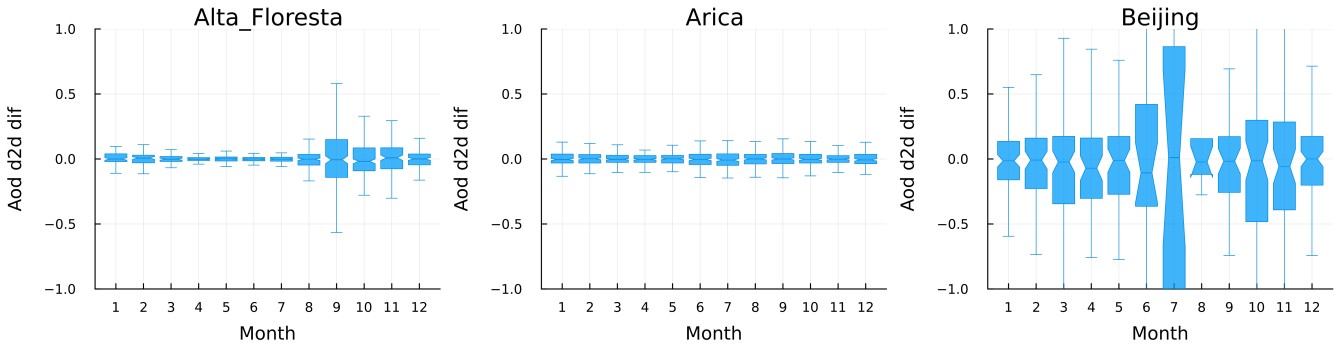

**Figure 4.** Monthly distribution of the absolute day-to-day difference in daily mean AOD for three selected sites, representing seasonal variability, small and large intra-annual variability. The vertical range is homogenized to be [-1, 1].



## 4.2 AOD Forecasts

In terms of which method forecasts day-ahead AOD the best, Table 3 summarizes the best-performing forecasting methods for each site based on Pearson correlation coefficient, RMSE or MAE. At ten of 21 sites, CAMS forecasts perform the best with the maximum correlation and the minimum errors. Four sites have the highest correlation and lowest errors by the AERONET persistence method. Based on the minimization of RMSE, AERONET monthly mean performs the best for five sites, although the correlation and MAE would suggest using AERONET persistence at four of these five sites. In Mexico City, the highest correlation can be achieved by persistence, whereas using the AERONET monthly mean leads to the smallest errors. The persistence also has good performance in minimizing MAE in both sites Lampedusa and Tamanrasset, but CAMS forecasts are more successful in terms of correlation and RMSE.

**Table 3.** 21 AERONET sites, corresponding aerosol types, mean AOD and the best-performing forecasting methods (AERONET persistence is denoted as p, AERONET monthly mean as m, and CAMS forecasts as c) for each site based on maximum Pearson correlation coefficient (corr) or minimum errors (RMSE or MAE). Sites marked with * are classified with more than one typical aerosol type (Table 1).

| Aerosol type | Site | mean AOD | max corr | min RMSE | min MAE |
|---|---|---|---|---|---|
| Biomass | Alta Floresta | 0.29 | p | p | p |
| | Buenos Aires* | 0.33 | c | c | c |
| | Lake Argyle | 0.12 | p | p | p |
| | Mongu | 0.28 | p | p | p |
| Urban-industrial | Arica* | 0.19 | p | m | p |
| | Belsk | 0.21 | c | c | c |
| | GSFC | 0.15 | c | c | c |
| | Lille | 0.18 | c | c | c |
| | Mexico City* | 0.34 | p | m | m |
| | São Paulo* | 0.21 | c | c | c |
| | Thessaloniki | 0.21 | c | c | c |
| Mixed | Bandung | 0.45 | p | m | p |
| | Beijing | 0.57 | c | c | c |
| | Kanpur* | 0.70 | p | m | p |
| | Osaka | 0.26 | c | c | c |
| Dust | Banizoumbou | 0.48 | c | c | c |
| | Capo Verde | 0.12 | p | m | p |
| | Kuwait | 0.37 | p | m | p |
| | Tamanrasset | 0.26 | c | c | c |
| Maritime | Lampedusa* | 0.17 | c | c | p |
| | Santa Cruz Tenerife* | 0.15 | c | c | c |





If grouped by aerosol types, at three biomass aerosol sites, the persistence method has the advantage. On the other hand, CAMS forecasts perform the best for the two maritime sites (both also partly loaded with dust aerosols). The performance is ambivalent at dust aerosol sites: two of the four sites favor AERONET (persistence or monthly mean), and the other two sites obtained better results from CAMS forecasts. As for the urban-industrial sites, which are the most numerous in our analysis,

CAMS forecasts support more sites (5) than the AERONET methods (2).

In the following, the three accuracy measures are examined individually for each forecast method. Figure 5 shows the correlation coefficients of the AERONET measurement with the AOD forecast by three forecast methods detailed earlier. Based on the correlation, CAMS forecasts perform the best at 12 of the 21 sites, and the second best at 5 other sites, thus generally outperforming the forecast methods using AERONET AOD. The correlation can be as high as nearly 85 % by AERONET

persistence at several sites to as low as < 15 % by CAMS forecast in Mexico City. For one site, such as Arica, forecasts by these three methods can differ a lot or be fairly close, such as in Kanpur. CAMS forecasts perform the worst among the forecast methods at the following sites: Mexico City, Kuwait and Capo Verde. Furthermore, Mexico City and Bandung are sites where all the forecast methods fail to achieve a correlation coefficient higher than 0.5. Hamill et al. (2016) pointed out that Mexico City is one of the most difficult sites to classify since besides urban-industrial aerosols, biomass and mixed aerosols are almost

equal-proportionally present there. Besides, Mexico City is a site that is advised to exclude due to volcanic eruptions when calculating the global mean using CAMS reanalysis (Inness et al., 2019).

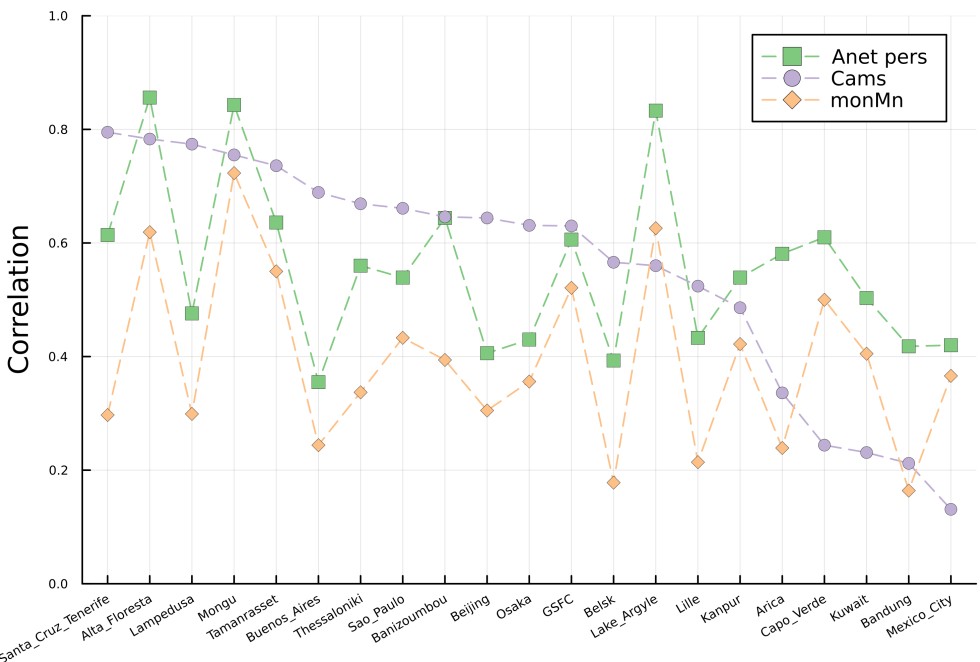

**Figure 5.** Correlation coefficients of the AERONET measurement with the AOD forecast by three forecast methods: AERONET persistence (green rectangles), CAMS forecast (purple circles) and AERONET monthly mean (orange diamonds), sorted in descending order by CAMS forecast.



Figure 6 shows the MAE (top) and RMSE (bottom) of the AOD forecast by three forecast methods compared with the AERONET measurements. Arica and Buenos Aires are sites with the sites with the lowest errors. On the contrary, Beijing and Kanpur are among the sites with the highest errors. At most sites, MAE and RMSE in AOD forecasts are close using three forecasting methods; exceptions are Kuwait and Mexico City, where CAMS forecasts produce much larger errors than using
AERONET-based forecasting methods. With smaller differences in errors, CAMS forecasts also perform the worst among the three forecast methods at the sites Bandung and Capo Verde.

At the sites Lampedusa and Tamanrasset, both loaded with dust aerosols, despite the discrepancy of forecast methods based on minimum RMSE or MAE, the MAE using the AERONET persistence does not differ much from CAMS, which supports the use of CAMS as the optimal forecast method. At the sites Bandung, Capo Verde, Kanpur and Kuwait, where the minimum
MAE and minimum RMSE indicate distinct optimal forecast methods, the lower panel of Fig. 6 reveals that the RMSE by AERONET monthly mean is very close to that by the persistence.



**Figure 6.** MAE (top) and RMSE (bottom) of the AOD forecast by three forecast methods: AERONET persistence (green rectangles), CAMS forecast (purple circles) and AERONET monthly mean (orange diamonds), sorted in ascending order by CAMS forecast.



## 4.3 DNI forecasts

To account for the diurnal AOD variability, Table 4 shows the accuracy measures of DNI using daily or hourly AOD from
CAMS for the site Beijing. The correlation of DNI using daily AOD from CAMS with daily AOD from AERONET measure-
ments is slightly higher than when hourly AOD is used. However, using hourly CAMS AOD leads to smaller errors in DNI
than daily values (the difference being as low as 5 %). However, note that the hourly AOD measurements at AERONET sites
are limited and irregular, resulting in few coincident data points with the hourly AOD by CAMS. Thus, the comparison of
hourly AOD is based on much fewer data points than using interpolated daily AOD.

**Table 4.** Comparison of accuracy measures of DNI (corr unitless, the other measures in $\mathrm{Wm}^{-2}$) using daily or hourly AOD from CAMS for
the site Beijing.

| Stats | Daily | Hourly |
|---|---|---|
| corr | 0.847 | 0.824 |
| RMSE | 168.7 | 160.5 |
| MAE | 124.5 | 118.8 |
| MBE $\pm$ std | -60.4 $\pm$ 157.5 | -42.2 $\pm$ 154.9 |

Figure 7 presents an example of the day-to-day AOD variation versus the relative deviation of DNI forecasts for the site Thes-
saloniki from July 2015 to December 2020 based on AERONET persistence or CAMS forecasts with AERONET measurement.
Both forecasts reveal a negative relationship between day-to-day AOD variation and relative deviation in DNI forecasts, with
the distinction that AERONET persistence also forecasts sporadic data pairs with a positive correlation.

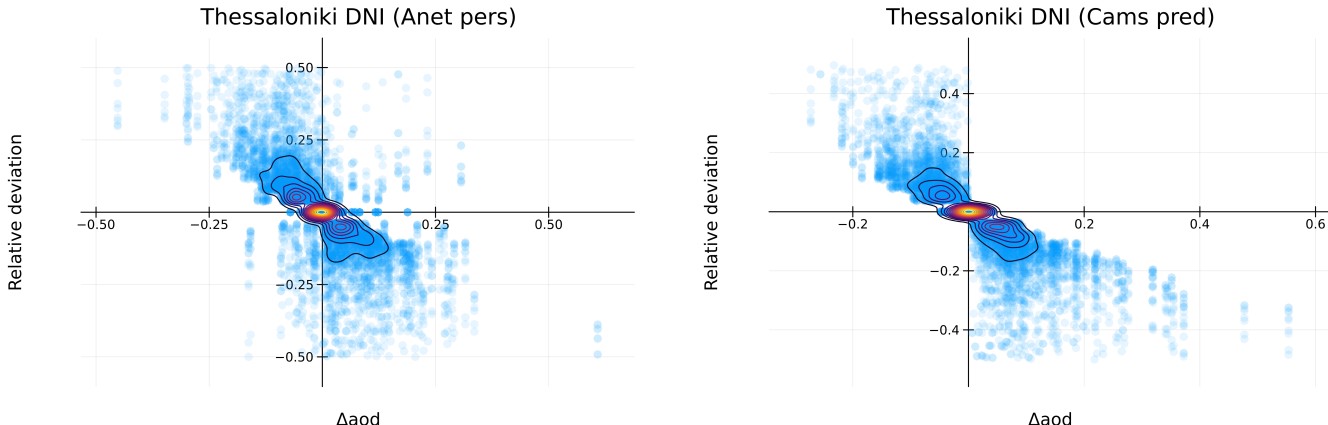

**Figure 7.** Relative deviation of DNI forecasts versus day-to-day AOD variation in Thessaloniki from July 2015 to December 2020 based on
AERONET persistence (left) or CAMS forecast (right) with AERONET measurement.





When all selected sites are considered, we can find a positive relationship between the relative RMSE of DNI forecasts based on AERONET persistence and mean absolute day-to-day AOD variation, as shown in Fig. 8. Color codes denote the mean AOD

of each site. The majority of these sites have a mean AOD below 0.4. The mean absolute day-to-day AOD variation at most sites is below 0.2, corresponding relative RMSE lower than 30 %. Beijing has a slightly lower mean AOD than Kanpur, yet the mean absolute day-to-day AOD variation in Beijing is much higher than other sites, which results in a relative RMSE in DNI much higher, reaching > 50 %. On the other hand, it can be confirmed again that Arica, as one of the sites with the smallest day-to-day AOD variation, experiences the smallest relative errors in DNI forecasts using AOD by AERONET persistence.

Empirically, more than two-thirds (15/21) of the sites exhibit a mean absolute day-to-day variation in AOD within 30-50 % of their mean AOD.

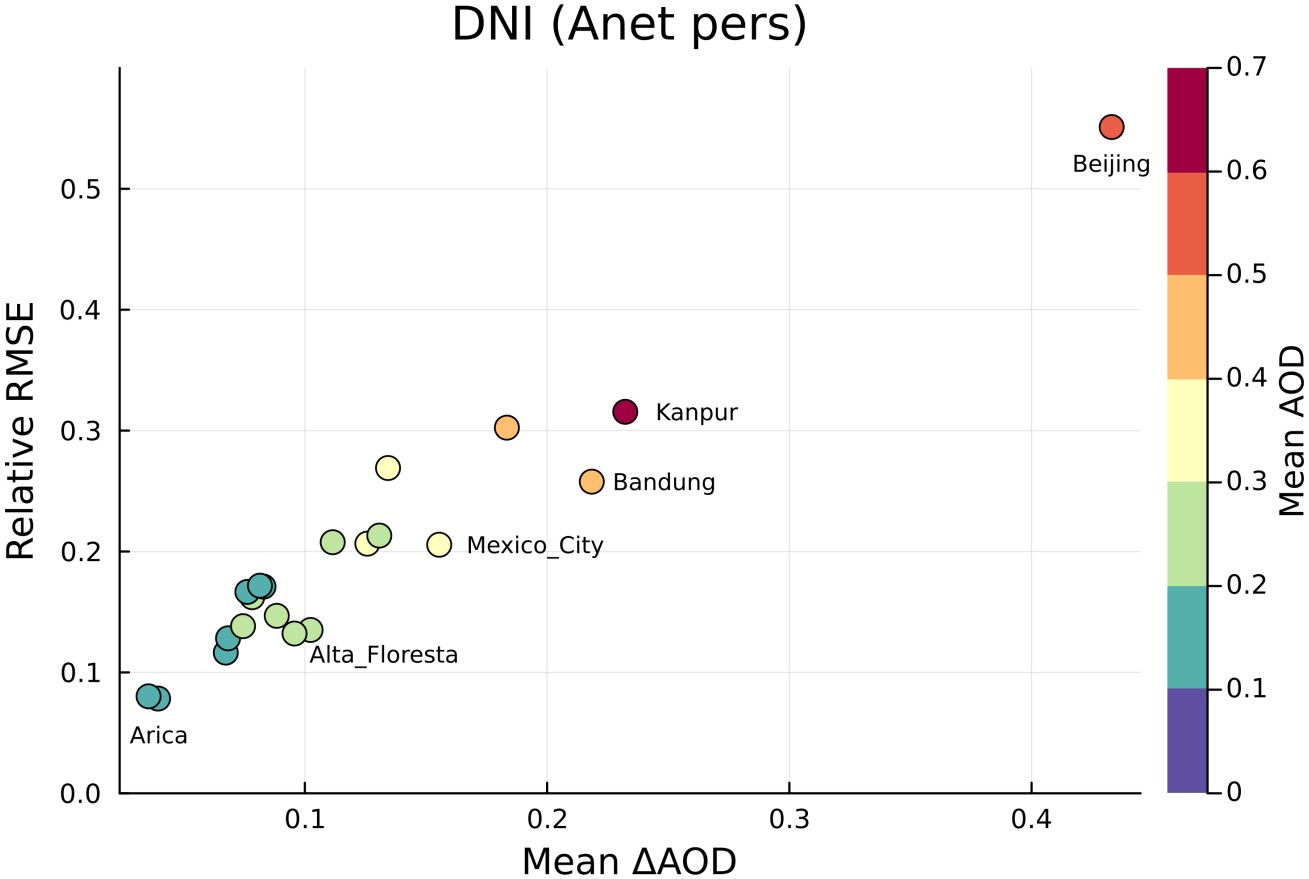

**Figure 8.** Relative deviation of DNI forecasts versus mean day-to-day AOD variation by AERONET persistence for 21 sites. Color codes denote the mean AOD of each site.

To summarize the performance of each AOD forecast method in day-ahead DNI forecasting, Fig. 9 presents the percentage of days at each site with DNI deviation > 5 % and Fig. 10 with the percentage > 20% due to day-to-day AOD variation using



AERONET persistence, CAMS forecasts and AERONET climatology. For most sites, when the threshold is set to 5 %, more
than 60 % of the days (up to 100 %), the DNI deviation is higher than this threshold, regardless of the forecast method used
for AOD. If a DNI deviation of within 20 % is chosen, most sites have at least half of the days satisfying this criterion (10–50
% of the days failing), notably the southern American site Arica (< 10 % of the days with > 20 % DNI deviation) and all four
European sites (< 20 % of the days). Exceptions include Beijing, which would have more than 50 % of the days with DNI
deviation > 20 % using any of the three forecast methods for AOD. The site Kuwait in the Middle East would also experience
60-70 % of the days with higher than 20 % deviation in DNI forecasts when CAMS AOD forecast is used; the percentage
of such days would decrease to < 40 %, when adopting forecast methods from AERONET (persistence or monthly mean).
Kosmopoulos et al. (2017) pointed out that CAMS overestimates DNI under high aerosol loads, which to a certain extent
explains the inferior performance of CAMS forecast for the sites Beijing and Mexico City, where there are predominantly
mixed aerosols. Another location to take caution is the northwestern African site Banizoumbou (situated south of the Saharan
desert) since all three forecast methods report ±50 % of days surpassing the DNI deviation threshold of 20 %, which indicates
less reliable forecasts there than at fellow dust aerosol sites. In the end, an acceptable deviation in DNI depends on the location-
specific requirements of user groups.





**Figure 9.** Percentage of days with DNI deviation > 5 % due to day-to-day AOD variation using AERONET persistence (top), CAMS forecasts (middle) and AERONET climatology (bottom).



**Figure 10.** Percentage of days with DNI deviation > 20 % due to day-to-day AOD variation using AERONET persistence (top), CAMS forecasts (middle) and AERONET climatology (bottom).



Last but not least, Figure 11 shows the relative RMSE (rRMSE) in DNI forecasts due to day-to-day AOD variation using AERONET persistence, CAMS forecasts and AERONET climatology. The relative RMSE in DNI at most sites is lower than 20

%. The sites Arica and Lake Argyle have the minimum rRMSE (< 10 %), and Beijing has the maximum, by both AERONET persistence and monthly mean. Using CAMS forecast, the site Capo Verde has the lowest rRMSE and Kuwait the highest. The sites Beijing, Mexico City, and Buenos Aires could expect improvements in the CAMS AOD forecast to reduce the deviations in DNI forecasts there.

## 5 Summary and outlook

To sum up, this study analyzes the spatiotemporal variability in AOD from ground measurements. CAMS AOD forecast is compared with forecast methods based on AERONET measurements. The induced deviation in DNI forecasts due to day-to-day AOD variation is also quantified, and implications in terms of geographical regions as well as aerosol types are derived. Day-to-day AOD variability is high at locations with high aerosol load, e.g., Beijing and Mexico City, both characterized by mixed aerosols. At dust aerosol sites, we also found high day-to-day AOD variability.

At different sites, the optimal AOD forecasts with the highest correlation or the smallest errors come from different data sources and forecast methods, which the sites' representative aerosol types can sometimes inform, providing information about the usability of model-based AOD forecasts as alternatives to AOD forecasts using ground measurements. CAMS forecasts perform better at more sites than AERONET persistence, among them many urban-industrial aerosol sites. AERONET persistence forecasts AOD with lower errors at dust aerosol sites. Under cloudless conditions, AOD variability results in the deviation

of DNI forecasts from actual values, which demonstrates the relevance of AOD accuracy to DNI forecasts and the monitoring and management of CSP systems. At the accuracy level of 5 % deviation in day-ahead DNI forecasts, none of the AOD forecast methods discussed here satisfactorily meet the requirements. Yet, we can expect better results achievable at many more sites with a threshold of 20 % DNI deviation.

For prospect research, seasonal and interannual variability or trends of AOD could be examined. Relative deviations in hourly

DNI caused by deviations in hourly AOD forecast could be quantified and compared with clear-sky climatology. Moreover, to corroborate or elaborate on the findings about the usability of model-based AOD forecasts or forecasts based on ground measurements presented here, more site-specific case studies are needed. One can further investigate the characteristics of SSR forecasts on locations with different aerosol types. In addition, research in this field would benefit from longer quality-assured surface-based aerosol measurements.

*Code and data availability.* Version 3 AOD data are freely available from the AERONET website (https://aeronet.gsfc.nasa.gov, last access: 1 December 2024). All the used and processed data for this paper can be requested from the corresponding author.





**Figure 11.** Relative RMSE in DNI forecasts due to day-to-day AOD variation using AERONET persistence (top), CAMS forecasts (middle) and AERONET climatology (bottom).



*Author contributions.* Idea and initialization: XH, SK; data provision and curation: XH, KP; first draft writing, visualization, analysis, and interpretation: XH; writing, review, and editing: all authors.

*Competing interests.* The authors declare that they have no conflict of interest.

*Acknowledgements.* This research was partly funded by the "EXCELSIOR": ERATOSTHENES: Excellence Research Centre for Earth Surveillance and Space-Based Monitoring of the Environment H2020 Widespread Teaming project (www.excelsior2020.eu). The "EXCELSIOR" project has received funding from the European Union's Horizon 2020 research and innovation program under grant agreement No. 857510 from the Government of the Republic of Cyprus through the "Directorate General for the European Programmes, Coordination, and Development" and the Cyprus University of Technology. Part of this work was supported by the COST Action Harmonia (CA21119)
supported by COST (European Cooperation in Science and Technology). We thank the teams of the AERONET for ground measurements and maintenance, and CAMS for the data production and distribution.



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
