# Peer review of "Assessment of aerosol optical depth forecast for day-ahead clear-sky direct irradiance"

_EGUsphere, 2025_

## Author Response (AR1)

The authors thank the reviewers for their comments and suggestions. The point-to-point replies to the reviewers' comments and questions are listed below following >.

**Referee 1:**

**General comments**

Overall, this is a very nice study, looking into the day ahead forecast accuracy of AOD and DNI for solar energy production in cloud-free conditions. It is re-doing the study of Schroedter-Homscheidt et al. (2013 and 2016), but now for the recent CAMS forecast generation instead of the previously used AERONET-only assessment for cloud-free conditions (2013 paper) and for the precursor dataset from MACC but for all-sky conditions (2016 paper, DOI 10.1127/metz/2016/0676).

Furthermore, additional statistics are provided, but on the other hand the intra-day variability is assumed to be neglectable and only daily means of AOD or daily sums of DNI are compared. This is counter-intuitive as power markets require hourly resolved forecasts or even 15 min resolved forecasts. It may be very well justified by AOD variability, but requires clearer justification to serve reader/users requirements in the solar energy sector.

> The main goal of the paper was to investigate day-to-day AOD variability based on long-term AERONET series of consecutive days and to explore the model's ability to capture this variability. When it comes to practical aspects, the energy market deals with different time horizons: intraday, one day ahead, two days ahead etc.

In addition to the scientific question on day-to-day variability for different areas and aerosol types, we tried to address the reviewer's comment related to the energy market:

- Intraday: This was not the scope of the paper, but for sure it plays a role in using averages on days when AOD is variable, the way we dealt with this was to make a sensitivity analysis on the use of one daily value based on the fact that also intraday AOD is variable (Section Results). At the beginning of Section Methodology, we have cited the analyses of DNI deviation due to intra-day AOD variation by Schroedter-Homscheidt et al. (2013) again before explaining that we used daily mean AOD.
- 2-day forecast: We agree with the reviewer that practically next day forecasts require the previous day AOD. For most of the AERONET stations this is available in real time or near real. However, AOD average can be retrieved only at the end of the day so this has practical implications (timewise) in market forecasts for the next day.

So we decided, after the reviewer's recommendation, to include the 2-day forecast analysis in addition to the 1-day in the revised paper.

Furthermore, power markets require day ahead forecasts to be provided in the late morning (e.g. 10 or 11 am). Therefore, a 1-day persistence as evaluated here (FC method 1) is not of relevance for the solar energy sector in day-ahead-markets, it only may be of relevance for intra-day-markets for the afternoon hours (where there is the smallest forecast value). Only the AERONET observations from yesterday can be used in the persistence approach for the

day-ahead due to the timing constraints of energy markets. The typical persistence approach as used in meteorology is not applicable in the solar energy community. To make the study relevant for the solar energy sector in day-ahead markets, this '2-day-persistence' approach needs to be added or the current persistence approach results need to be replaced by the '2-day-persistence'.

Especially, the 2nd point requires a major revision in the sense of re-running the results.

> We have added the 2-day persistence results, replacing, or for some figures/analysis in addition to, the 1-day persistence. More details see replies below.

Furthermore, I would like to suggest to think about a more specific title for the paper. The current title is very general and a bit misleading. The paper is more oriented on assessing the forecast accuracy of CAMS AOD forecasts versus AERONET-based persistence and climatology as naïve forecasts.

> We changed the title to "Assessment of aerosol optical depth forecast for day-ahead clear-sky direct irradiance".

Comment to the editor: The scope of ACP may fit a bit better than AMT, but I understand that this is a manuscript for a joint special issue of ACP and AMT.

**Specific comments**

Abstract, line 11/12: Please specify the term '5% deviation' better already at the beginning. Do you mean in daily means of AOD/daily sums of DNI or in hourly day ahead DNI forecasts? This should be clear already when reading only the abstract.

> 5% deviation refers to daily sum of DNI [Wh m$^{-2}$] derived by AOD forecasts compared to by ground AOD measurement. We have clarified this by using "direct normal irradiation (daily sum)" in the abstract.

Also, after reading the introduction, it is still not defined what temporal resolution you look at in the study. This only gets clear in the method section, but also this section starts with an unjustified assumption of no intra-day variability. Is there any knowledge available in the literature to justify this assumption via a citation?

> We have specified the daily resolution we used in Section Methodology. Please also refer to the reply to comment one.

Later, a sensitivity analysis of this assumption is done for the only site Beijing. I wonder if this is a good choice. Intra-day variability of AOD can be expected e.g. in areas with dust fronts in deserts. I'm not sure if Beijing with its urban AOD sources is a very good place to expect intra-day variability. It may be wise to repeat this assessment for more stations with more event-like meteorological phenomena?

> We have added other two sites with low and moderate AOD variability (Lake Argyle and Thessaloniki) in the sensitivity analysis. Please refer to the new Section 4.3 Daily and hourly AOD.

Table 4. Comparison of accuracy measures of DNI (corr unitless, the other measures in Wm−2) using daily or hourly AOD from CAMS for the sites Beijing, Lake Argyle and Thessaloniki.

|  | Daily | Hourly |
|---|---|---|
| **Beijing:** | | |
| corr | 0.678 | 0.298 |
| RMSE | 242 | 340 |
| MAE | 183 | 275 |
| MBE ± std | -92.5 ± 224 | -116 ± 319 |
| | | |
| **Lake Argyle:** | | |
| corr | 0.955 | 0.855 |
| RMSE | 90.0 | 115 |
| MAE | 63.2 | 84.5 |
| MBE ± std | -46.9 ± 76.7 | -32 ± 110 |
| | | |
| **Thessaloniki:** | | |
| corr | 0.957 | 0.719 |
| RMSE | 73.4 | 146 |
| MAE | 53.0 | 112 |
| MBE ± std | -13.6 ± 72.1 | -17.8 ± 145 |

Furthermore, in table 4 it is not clear if these are statistics in hourly temporal resolution or again in daily resolution but with hourly input (which would be not relevant for the reader). For users it is relevant if the hourly DNI is predicted well, daily DNI is of no interest in the forecasting of solar energy as there are (at least to my knowledge) no energy markets working on daily sums of tomorrow. Therefore, it would be needed to prove that your approach of looking into daily data is of value for the solar energy community with their hourly DNI forecast requirement. I'm not saying that your assumption is wrong, only it is not clearly discussed and justified to the users.

> Results in Table 4 is for hourly DNI forecast and we have clarified this in the text and updated table caption.

Also, I would spend an extra sub-chapter with a clear and easy-to-find heading on the question if daily assessment is sufficient. Knowing that power markets need at least hourly resolved radiation day ahead forecasts, the gap between your results and the user community needs to be filled with a good and (for the reader) easy to find explanation. Currently this part is very much split and therefore hidden separate statement in various chapters.

> We have separated the section for comparing the effects of daily and hourly AOD on DNI forecasts.

Introduction:

Line 14 onwards: Authors may also want to discuss the relevance of DNI for any tracked system, e.g. tracked PV or the correct diffuse/direct split needed for Agri-PV with their various tilted surface options.

> We have briefly mentioned the relevance of direct irradiance also for tracked PV by citation(s).

The paper correctly discusses two impacts of aerosols on DNI, the extinction and the soiling of surfaces. I'd appreciate if the references in line 20 onwards would be better sorted in references on the extinction and references of the soiling effect. It is ok to name the soiling issue, but as the paper does not deal with this further, the two groups of references should not be mixed up as done at the moment.

> We have reorganized the soiling references by putting them at the end of the paragraph, while keeping them short, since they are not the focus of this paper.

It may be worth to ensure that the reader understands that all DNI comparisons assume cloud-free conditions during the whole day. This study design is ok for an assessment of aerosols, but solar energy community readers may not be aware of this constraint of the study. This should be mentioned very clearly in abstract and introduction, that cloudy cases are not treated in the assessment. Therefore, all results are not to misinterpreted as 'all-sky forecast accuracy' which users will search for.

> We have used the keyword "cloud-free" in the abstract and intro.

It may be very helpful for the reader to understand in the introduction, that this study is revisiting work by Schroedter-Homscheidt et al (2013 and 2016) with a similar approach and metrics as suggested by them, but for more recent CAMS AOD forecast cycles. There is no doubt, that an assessment of more recent CAMS AOD forecasts after more than 10 years is a very valuable scientific contribution, and it is welcome to see that the authors are adding further detailed assessment at some aspects/locations. Nevertheless, it may be wise to cite both papers for the reader's orientation.

> We have specified the revisiting aspect at the end of the intro.

Data section:

CAMS forecasts and CAMS reanalysis are run with different software versions. Strictly speaking, citing a reanalysis assessment is not a proper reference without further description of the differences of both products. Regular CAMS forecast assessment reports may be a good alternative.

> We have changed the reference for CAMS forecast to be the latest validation report Bouarar et al. (2024).

Method section:

As discussed above: Please add a '2-day persistence' for the day-ahead application case or replace the '1-day persistence' (which is only valid for the intra-day application case) by the

'2-day persistence' to fulfill your goal of assessing day-ahead forecast capability. All assessments need to be re-run to either show the 2-day persistence (as the one needed primarily by the solar community) or to show both persistence approaches.

> We have added the 2-day persistence approach. Please note the updated figures from Fig. 5 onwards.

Results:

Table 3 is strange. As a reader I want to know the performance quantitatively. I want to decide if differences are significant. Why do you only give the method name which performs best? Why is this table not quantitative?

Are the quantitative results given later in fig. 5 and 6? If yes, then table 3 is superfluous perhaps? You may want to order the stations in Fig 5 to 6 in the groups you introduce in table 3 and then delete table 3?

> Table 3 shows the best performing method for each site for an overview. The quantitative results are given in Figs. 5 and 6 for an easier comparison. Resultes in Figs. 5 and 6 are sorted in descending order by the performance of CAMS forecast, which provides a clear orientation.

Talking about grouping of stations, you already introduced a grouping in Fig 2 and 3 according to geographical area. Perhaps you want to use only the grouping introduced in table 3 in the whole paper as it is the aerosol type related grouping (which has more physical meaning than a pure geographical area)?

> We have adopted the grouping by aerosol types and moved the coordinates of the sites in Table A1 to the appendix.

What is the value of Fig. 7? Isn't this just the expected behavior? Please tell the message of this plot better, or omit it?

> In Fig. 7, there is a concentration of data points at the origin in both plots. However, the distribution of relative DNI deviation also differs using AERONET persistence or CAMS forecast.

Also fig. 8 looks rather as expected. In case of no variation from day to day, the persistence will do the job. What is the 'news' in this plot? Why is it worth to show this plot? Can you elaborate on that?

> Fig. 8 quantifies the linear relationship between mean difference in daily AOD and relative RMSE in DNI, which aids in estimating the DNI deviation for further sites once the mean difference in AOD is known. We have specified this in the manuscript. Of course low day-to-day variability will indicate good persistence related results. However, one of the added values of this figure and the paper is to point out the areas/ stations around the globe and different aerosol types for which this is especially true.

**Technical corrections**

Abstract: Sentence 'AERONET persistence…', line 10 seems to be incomplete.

> We have added "approaches do" at the end.

Line 66 & table 2: 'what is meant by are available from 2015'. CAMS ADS catalogue entry states: "EAC4 is only available from 2003 onwards." Why do you state 'since 2015'?

> EAC4 being a reanalysis dataset indeed starts from 2003. However, the dataset CAMS global atmospheric composition forecasts starts only from 2015 (https://ads.atmosphere.copernicus.eu/datasets/cams-global-atmospheric-composition-forecasts?tab=overview)

Fig 5 and 6: This is a daily mean AOD forecast? Perhaps add this?

> Yes, we have added "daily mean AOD" to the figure captions.

Fig 8: what is relative deviation of DNI forecasts? Which unit? How is it normalized (mean of all values, mean of all daytime values, if the latter - how is daytime defined)?

> Relative RMSE here is unitless and lies in the range [0, 1]. It is normalized by the individual mean of all daytime values at each site. We took daylight values from CAMS forecast from 6 am to 6 pm locally, then the CAMS AOD values are synchronized with AERONET measurement time stamps.

Fig 9/10/11: better clarify that DNI is daily sum of DNI deviation due to daily mean AOD variation ?

> We have clarified the caption to be "Percentage of days with > 5 % deviation in daily sum of DNI due to day-to-day AOD variation …"

**Referee 2:**

This work compares the use of aerosol data from AERONET and CAMS for the day-ahead forecast of direct normal irradiance (DNI). Three forecast methods of AOD are used together with a radiative transfer model: a monthly mean and the next day persistence method based on AERONET surface data and the day-ahead forecasts based on CAMS products. The paper is well organized and the topic is worthy of investigation. However, there are some aspects that should be additionally considered/clarified.

1. The work compared the use of AERONET and CAMS data for the day-ahead forecast of DNI, taking as a reference the AERONET data for that day. The use of other ground-based measurements of DNI, for example, from reference stations collocated to AERONET sites, is not mentioned (neither in the abstract nor in the data section), and this should become clearer. This must be clarified in the abstract and in the introduction and possible deviations in relation to reference DNI measurements must be also mentioned. Furthermore, the abstract starts by referring that the spatiotemporal variations of AOD at a global level are examined, and only later does it become clear that one of main objectives is to compare different data/method combos for day-ahead forecasts of DNI.

> We have clarified in the abstract that we are not using DNI measurements but only simulations ("The performance of each AOD forecast method in DNI forecast is assessed and compared"). We also agree that the day-ahead (and in the revised manuscript 2-day ahead) AOD forecast accuracy is the main concept and not global spatiotemporal variations. However, global day-to-day variations observed for various locations worldwide (due to the different aerosol sources and transport processes) provides a very good hint on how well the forecast methods can work in the different locations.

In addition, the absence of comparing DNI forecasts with actual DNI measurements has to do with the fact that we aim for estimating the AOD forecast effects on DNI and not to include other sources of uncertainties such as model to measurement agreement or dataset limitations due to the need of consecutive "cloudless days" (100% of cloudless data within a day).

So we think that the right approach was to use real AOD and AOD forecasts in the same model and calculate the DNI difference for 100% cloudless days, in other words only the effect of aerosols in the DNI forecast.

2. The assumption that AOD is invariant during the day must be further discussed and justified and that it can be calculated from three records only. To what extent is this assumption valid in view of the deviations observed in the persistence method under evaluation? Or, how this assumption can affect (positively or negatively) the performance of the forecast methods, namely the persistence method? Also, please clarify if any interpolation method is used to obtain the AOD forecasts from CAMS for the AERONET sites (L.81).

> We agree with the reviewer that the AOD is not invariant during the day. This is a problem in many studies using AOD from AERONET data as they are not continuous due to the fact that sun photometers cannot measure during periods when the sun is covered by clouds.

We did a sensitivity study on the effect of temporal resolution for the site with the highest (worst case scenario) variability: Beijing.

Theoretically, using few measurements per day should have an effect only if various systematic aspects happen, meaning a combination of a systematic AOD daily pattern accompanied by a systematic cloud presence. Otherwise e.g. a day with AOD increase will not lead to an important systematic DNI shift due to the average AOD use.

In order to investigate this more we used hourly CAMS data for 3 sites with high, low and moderate day-to-day AOD variability: Beijing, Lake Argyle and Thessaloniki.

Average AOD (550nm) based on all hourly CAMS values for daylight hours AOD_c

Average AOD based on all coincident with AERONET, CAMS values AOD_c_a

| Site | AOD_c | AOD_c_a |
|------|-------|---------|
| Beijing | 0.685 | 0.698 |
| Lake Argyle | 0.129 | 0.131 |
| Thessaloniki | 0.193 | 0.193 |

In the following figures we compared the AOD differences distribution for each day and for the three sites:

[Figure]

Figure: percentage distribution of cases of the AOD difference for 3 sites: mn1 (CAMS daily average AERONET coincident data) - mn2 (CAMS daily average all data).

Finally, CAMS and AERONET comparisons are based on daily integrals. Matching CAMS to AERONET data included the closest hour value from AERONET to the (hourly available) CAMS value.

3. How exactly the daily values/simulations of DNI are determined (e.g. how many values are used), for example to produce the relative deviation graphs of DNI in Figure 7? How these values are representative of the true daily DNI values when only three AERONET records are available?

> The DNI was determined based on model simulations using the average daily AOD value and calculating hourly DNI for the daylight hours for each specific day/location. So for Figure 7 and onwards, daily DNI comparisons are concerned. As clarified in the previous response, the fewer available AERONET measurements, the more uncertainty would be introduced in DNI calculations.

However, the study is mostly investigating the effects of possible AOD forecast methods on DNI forecasts including the shortcoming of sunphotometric measurement-based methods that cannot measure continuously. To be more specific, the paper describes only the contribution of aerosol forecast in possible DNI forecast models and cannot capture days with partly cloudiness. Only including consecutive 2-3 cloudfree days would limit the analyzed dataset.

As clarified in the reply to Comment 1, we try to separate the problem of simulating DNI accurately with model calculations (that is based on input availability, model performance and DNI measurement availability) with the problem of AOD forecast accuracy and effects on a one- and two-day ahead forecast of DNI assuming that these next days are cloudfree.

For having such an analysis in order of priority we could have: continuous measurements under cloudfree conditions, continuous AOD measurements/model inputs, at least few AOD measurements per day, one daily satellite measurement. Depending on the use (global, local, temporal resolution) needed, different approaches have their advantages and disadvantages. In our case, we based our study on existing (hourly CAMS and AERONET random during the day) AOD data on a global scale, using them as model inputs for DNI calculation.

4. The sentence starting "Pre-calculated look-up tables …" (L.98-100) is not clear. Please clarify if a complete batch of simulations was carried out for generating this tables for the indicated ranges and then deviations were determined. If so, which interpolation method was used?

> We cited Papachristpoulou et al. (2022), which described the complete batch of simulations and interpolation in detail: The frequency of RTM simulations was chosen to be hourly to account for the sun elevation. For the 1h RTM simulations, daily AOD values were assumed invariant in the day. They used the uvspec model to compute spectral irradiance forming bulk LUTs, which were integrated over the whole shortwave spectrum to obtain total irradiance. Interpolation was applied on the spectrally integrated irradiance to derive finer LUTs covering over millions of RTM runs as described in Papachristopoulou et al., 2022. Practically as shown in the later using the LUTs/interpolating or running libRadtran for each case has negligible differences.

5. In Table 1., a uniform resolution for lat. and lon. values may be used.

> We have adjusted the digits of the coordinates.

6. The construction of the graphs shown in Fig. 4 should be better explained, namely which reference value is used for plotting the deviations (positive or negative) of the absolute day-to-day differences.

> Day-to-day difference in AOD is based on the rolling day, e.g. day-to-day difference on Day 2 is daily mean AOD on Day 2 minus daily mean AOD on Day 1, etc. That is why there are both positive and negative differences in the figures.

7. The order of the AERONET sites on the horizontal axis of Figure 5 appears to be the one that produces a decreasing order of the correlation coefficient for CAMS. This criterion was not followed for the order in Figure 6. In the latter case, and in the absence of a table with values, using the same order in the top and bottom graphs would help to compare the RMSE and MAE values of the different sites.

> We have adjusted the site order based on RMSE for CAMS.

8. The analysis at the beginning of section 4.3 can be further improved (see also comment 2.). The way how the day-to-day variation in the number of valid AERONET records affects the metrics of the forecast methods should be further addressed. The number of valid AERONET records does not depend on the variation of the AOD.

> Please refer to the reply to Comment 2.

9. L15. "… in regions with high direct normal irradiance (DNI>200 W/m2) …" is not clear. Is this a filter to only consider records in which DNI under clear-sky conditions (already screened by AERONET) higher that 200 W/m2 are considered. Please clarify.

> This is a general threshold we cited from a previous related study. We did not use it to filter AERONET sites we include.

10. Other statistical indicators can be used in this analysis, which have been used more recently in similar works in this area, such as fractional bias, fractional gross error, global performance index and skill score.

> We have calculated the skill score of CAMS forecast in terms of RMSE compared with 2-day persistence using AERONET. It can be seen that the sites where CAMS forecast does not outperform AERONET persistence include: Arica, Kuwait, Lake Argyle and Mexico City.

| Site | FSS |
|------|-----|
| Alta Floresta | 0.11 |
| Arica | -0.09 |
| Bandung | 0.11 |
| Banizoumbou | 0.36 |
| Beijing | 0.18 |
| Belsk | 0.28 |
| Buenos Aires | 0.3 |

| | |
|---|---|
| Capo_Verde | 0.6 |
| GSFC | 0.27 |
| Kanpur | 0.24 |
| Kuwait | -0.46 |
| Lake_Argyle | -0.35 |
| Lampedusa | 0.25 |
| Lille | 0.31 |
| Mexico_City | -0.54 |
| Mongu | 0.19 |
| Osaka | 0.18 |
| Santa_Cruz_Tenerife | 0.42 |
| Sao_Paulo | 0.23 |
| Tamanrasset | 0.1 |
| Thessaloniki | 0.35 |